# “One and a Half Years of Things We Could Have Done”: Multi-Method Analysis of the Narratives of Adolescents with Type 1 Diabetes during the COVID-19 Pandemic

**DOI:** 10.3390/ijerph20032620

**Published:** 2023-02-01

**Authors:** Marta Tremolada, Maria Cusinato, Alessia D’Agnillo, Arianna Negri, Elena Righetto, Carlo Moretti

**Affiliations:** 1Department of Developmental Psychology and Socialization, University of Padova, 35131 Padua, Italy; 2Pediatric Diabetes Unit, Department of Women’s and Children’s Health, Padua University Hospital, 35128 Padua, Italy

**Keywords:** type 1 diabetes, COVID-19, adolescents’ narratives, psychological well-being, qualitative research, NVivo program

## Abstract

Background: Public health interventions for COVID-19 forced families to adopt changes in daily routines that affected children’s and adolescents’ psychological well-being. In youth with type 1 diabetes (T1D), psychological symptoms may compromise glycemic control and outcomes; however, evidence of improved glycemic control in children and adolescents with T1D emerged early during the pandemic. This qualitative study aims to provide a more in-depth understanding of how the COVID-19 pandemic affected adolescents’ with T1D routines, experiences, T1D management, and psychological well-being. Methods: 24 adolescents, aged 15–18 years, with T1D, joined focus-group discussions during the diabetes summer camp. Word frequency analysis and thematic analysis were conducted on adolescents’ narratives. The average frequencies of use of words related to COVID-19 and to T1D were compared by *t*-test. Results: Word frequency analysis identified “friends”, “family”, and “home” as the most recurrent terms. Seven themes were highlighted: (1) COVID-19 and T1D; (2) emotional reactions to the COVID-19 pandemic; (3) changes in daily life; (4) feelings of loss; (5) coping with the COVID-19 pandemic; (6) the COVID-19 pandemic as opportunity; (7) return to (new) normality. COVID-19 related words were on average more frequent than words referring to T1D. Conclusions: The COVID-19 pandemic may have represented a more stressful condition for adolescents with T1D, facing additional challenges compared to their healthy peers. Findings offer directions to the diabetes care team for customized interventions while the effects of the pandemic on adolescents’ health continue.

## 1. Introduction

Type 1 diabetes (T1D) is one of the most common chronic diseases in childhood and adolescence, with 1,211,900 youth (0–19) living with the disorder worldwide [1]. 

In type 1 diabetes, inadequate insulin secretion requires adherence to a lifesaving therapeutic regimen that includes blood glucose monitoring, multiple insulin administrations, carbohydrate counting, and regular physical activity [2]. Adequate glycemic control requires patients to perform strict daily self-management tasks responding to changes in activity, food, and physiology [3]. Daily T1D management could expose patients to increased stress, particularly during difficult transitions, such as adolescence. Adolescents with type 1 diabetes, while dealing with normal major physical, cognitive, emotional, and relational changes, assume increasing diabetes self-management responsibilities with emerging independence from caregivers. Pubertal insulin resistance, decreased adherence to treatment, decline in parental involvement, prioritization of social life, and risk-taking behaviors contribute to glycemic control deterioration commonly observed among adolescents with T1D [4]. Furthermore, eating, mood, and anxiety disorders, twice as common among adolescents with T1D as peers without diabetes, could compromise diabetes management, glycemic control, and quality of life [5].

Patient education is the key to successful type 1 diabetes management [3]. In pediatric patients, T1D management education includes the proposal of diabetes summer camps. Diabetes camps consist of a 7-day medically supervised stay, allowing children and adolescents with T1D to meet and share their experiences while learning autonomous diabetes care, enjoying recreational activities, and strengthening their relationship with the pediatric diabetes team [6].

During the COVID-19 pandemic, public health interventions against COVID-19, such as “stay at home” orders, school closures, and limited peer interactions, have taken their toll on adolescents’ psychological well-being, resulting in a consistent increase in reported symptoms of emotional distress [7,8,9,10]. In qualitative research having investigated adolescents’ experience during the COVID-19 outbreak, difficult emotions (e.g., sadness, anger, worry, irritability), home confinement, loss of milestones, and lack of peer interactions are common issues [11,12,13,14,15].

Children and adolescents with pre-existing chronic conditions, including youth with type 1 diabetes, were particularly vulnerable to experiencing psychological well-being deteriorations during the COVID-19 emergency [16].

At COVID-19 onset, adolescents with T1D faced additional challenges compared to their healthy peers, which may have negatively influenced their psychological well-being: while exposed to public discussion about the increased risk for a severe clinical course of COVID-19 in patients with diabetes, housebound adolescents with T1D needed to adjust their therapeutic regimen to profound changes in daily routines [17]. Concerns about possible detrimental effects on glycemic control have arisen.

However, early in the pandemic, evidence of improved glycemic outcomes emerged among children and adolescents with type 1 diabetes. Reduced physical activity but slowed-down routines, scheduled mealtimes, and more attention to self-care, with more regular insulin administration and frequent glycemic monitoring, could have contributed to better glycemic outcomes. Increased parental control and suppression of sports, school, and social occasions where adolescents struggle the most to adhere to T1D therapy could account for better diabetes-related outcomes. Regular consumption of homemade meals could have facilitated carbohydrate counting. The awareness that poorly controlled T1D could worsen the SARS-CoV-2 clinical course, as evident in the adult population, may have improved patients’ compliance with diabetes management [18,19,20,21].

The potential short- and long-term implications of the COVID-19 emergency on psychological well-being of adolescents with T1D remain partially unclear to date, with little evidence trying to deeply explain improved glycemic outcomes in children and adolescents more frequently emotionally distressed. Therefore, this study intended to explore the possible psychological and behavioral consequences of COVID-19 in this group, to understand adolescents’ perspective on evidence of improved glycemic outcomes, and to identify determinants of glycemic control during the COVID-19 outbreak. 

Specifically, our qualitative study aimed to expand on the current literature by providing a more in-depth understanding of how adolescents with T1D were doing and how the COVID-19 pandemic affected their routines, experiences, T1D management, and psychological health. For this purpose, we explored adolescents’ feelings and cognitions through their narratives.

## 2. Materials and Methods

### 2.1. Participants

We enrolled 24 adolescents (M/F = 17/7) aged 15–18 years (M = 16.75; SD = 1.07) with type 1 diabetes (mean time since diagnosis = 7.63 years; SD = 4.92; range = 1–16 years) followed up at the Pediatric Diabetes Unit at Padua University Hospital (Italy).

All participants were continuous/flash glucose monitor users, five were on continuous subcutaneous insulin infusion (CSII), and nineteen were on multiple daily insulin injections (MDI) therapy. Inclusion criteria were age between 15 and 18 years, duration of T1D of at least one year at enrollment, and voluntary participation in a 7-day diabetes summer camp. Exclusion criteria were comorbidity with psychiatric disorders, poor comprehension of the Italian language, and refusal to participate in diabetes camp activities.

Medical data on diabetes therapy and T1D duration were derived from clinical medical records. Table 1 describes the demographic and clinical data of the participants.

### 2.2. Procedures

Data collection took place between 26 July and 1 August 2021, in the context of the diabetes summer camp. At the time of data collection, the epidemiological risk of our country was considered low, and restrictions were considerably eased.

Campers participated in focus-group discussions. Focus groups typically consist of a specific topic-focused discussion among pre-selected participants with a shared background (e.g., adolescents with T1D during the COVID-19 pandemic) related to the research issue. The questions asked to the group by a trained moderator are carefully designed to stimulate discussion among participants. Focus groups aim to gain a more in-depth understanding of issues discussed from the perspectives of the participants. This makes focus groups an ideal method to explore topics about which little is known or to clarify unexpected results from previous quantitative investigations. 

Specifically, each participant joined two focus groups during the camp. Focus groups were simultaneously conducted by two psychologists. The focus groups’ script was discussed before the camp so that the moderators could address the same topics. Focus groups were recorded and transcribed verbatim. 

Written informed consent, including permission for audio recording, was obtained from the campers and their parents before data collection. Our investigation complied with the Declaration of Helsinki. This low-risk human research was part of ordinary clinical assistance activity. For these reasons, our study could be exempt from ethics review.

### 2.3. Data Analysis

Data were analyzed using a multi-method approach that included quantitative analysis (word frequency analysis) and qualitative analysis (thematic analysis). 

Most frequent words were checked through NVivo, a computer-assisted qualitative data analysis software that helps qualitative researchers collect, organize, analyze, visualize, and report the collected data [22].

After conducting preliminary word frequency analysis on integral transcripts, we checked the potentially ambiguous meaning of some of the 100 most frequent words (e.g., “good”/“well”), to exclude from the frequencies counting the number of cases in which words were used with meanings not significant or not relevant to our investigation. Furthermore, we separated the frequency of words that, still having the same linguistic form, had different meanings in the Italian language (e.g., “*sentire*” with both the meaning of “to feel” and “to hear on the phone”). The frequency of words with similar meanings was finally added up (e.g., “*amiche*” (female friends), “*amici*” (male friends)). 

We proceeded by creating two categories of words that were especially interesting for our research: terms referring to the COVID-19 pandemic (e.g., “lockdown”, “quarantine”, “pandemic”, “restrictions”) and words alluding to type 1 diabetes (e.g., “insulin”, “glycemia”, “hypo/hyperglycemia”). The average frequencies of use of words included in the “COVID-19” and “T1D” categories, respectively, were compared by *t*-test. The analysis was performed through IBM SPSS Statistics 28.

Reflexive thematic analysis of integral transcripts was performed using Braun and Clarke’s six-stage process [23], consistent with our interest in in-depth insight into the experience of adolescents. Two independent researchers conducted the thematic analysis focusing on what was personally meaningful to participants. Findings were interpreted within the context of the existing literature.

## 3. Results

### 3.1. Word Frequency Analysis

Word frequency analysis was carried out. The top ten most frequent words in adolescents’ narratives include “friends”, “home”, “before (COVID-19)”, “time”, “to feel”, “to meet”, “glycemia”, “together”, “great”, and “to miss”.

Specifically, “home” occurs one-quarter of the time together with the verbs “to stay”/“to be” or with the adjective “shut”. “Time” occurs in one out of three of the repetition with the meaning of “more time to”. The adjective “great” appears almost exclusively in the transcripts of the last focus groups, mostly about “return to normality”. “Family”, only 20th in frequency, reaches repetitions equal to “friends” by aggregating references to various family members (i.e., “mom”, “dad”, “parents”). “Friends” is, regardless of gender, the most frequently used word, followed by “home”. Compared to male participants, female adolescents utilize “glycemia”, “tough”, and “lonely” more frequently. Compared to female peers, male adolescents make more frequent use of the verb “to meet” and of words that refer to physical proximity (i.e., “physical contact”, “to hug”, “to be next to”). 

Figure 1 graphically illustrates the 100 most frequent words in adolescents’ narratives.

In our cohort, there is found a statistically significant difference in the average frequency of use of words related to COVID-19 (i.e., “lockdown”, “social distancing”, “to get sick with COVID-19”) and to “T1D” words (i.e., “glycemia”, “diabetes”, “insulin”) [t_23_ = 1.98, *p* < 0.05]. Specifically, words referring to COVID-19 (M_COVID-19_ = 6.33; SD_COVID-19_ = 4.04) are on average more frequent than words related to T1D (M_T1D_ = 4.34; SD_T1D_ = 5.61).

### 3.2. Thematic Analysis

Seven main themes were identified: (1) COVID-19 and T1D; (2) emotional reactions to the COVID-19 pandemic; (3) changes in daily life; (4) feelings of loss; (5) coping with the COVID-19 pandemic; (6) the COVID-19 pandemic as opportunity; (7) return to (new) normality.

Figure 2 graphically illustrates themes highlighted and relative connections. A complete description of the main and sub-theme, including participants’ most relevant and exhaustive quotes, is presented below.

#### 3.2.1. COVID-19 and T1D

This theme describes the impact of COVID-19-related changes in daily routines on T1D management. Implications of potentially belonging to a COVID-19 risk group are discussed. 

In our sample, 33% of participants reported improved glycemic control during the COVID-19 outbreak. Our data identify multiple aspects that contributed to improving diabetes-related metrics: (1) more time for self-care, with more regular blood glucose monitoring and insulin administration; (2) healthier home-cooked meals; (3) slowed-down, less unpredictable routines; (4) better sleep quality, with promptest glycemia corrections at night. In our sample, boredom increased blood glucose testing frequency. 


**
*R. (male, 16 years old):*
**
*[...] Glycemia slightly improved during the pandemic, I think because my routine has become much more monotonous. I was at home and had plenty of time to check my glycemia, even just because I was bored. Before the pandemic, I used to spend a lot of time outside, and honestly, I didn’t check my insulin pump very often, while at home, with a healthier diet and more time, my glycemia improved.*


Excessive attention to glycemic values, precipitated by the temporary need to self-manage T1D during the COVID-19 positivity period of her caregiver, is evident in the narratives of one participant.


**
*E. (female, 18 years old):*
**
*[...] I became obsessed with my metrics. In May, my mother got sick with COVID-19 and I had to start getting by myself. Glycemia has dropped. This made me happy, I admit, but I felt it was getting risky. [...] Now it annoys me to see even a 200 (glycemic value). I used to pay much attention to glycemic values even before, but now even more. I think being stuck at home didn’t help me.*


Worse glycemic outcomes were reported by 25% of our cohort and causally attributed to (1) restrictions to physical activity (sport, walks), (2) increased food intake, (3) continuous need to adapt T1D therapy, strongly sensitive to changes in routines, to different phases of the COVID-19 emergency, with variable restrictions in place and activities allowed, and (4) loss of motivation and difficulty continuing self-care and adhering to T1D therapy, given the circumstances.


**
*L. (male, 18 years old):*
**
*[...] At the COVID-19 onset, my routine changed completely. Glycemia went wrong because I was much more sedentary. I had to change my (therapy) parameters. During the summer period, there were many more activities allowed and I had to change my parameters again. Then we went back home. Now, the parameters have changed again! I think it’s true that having fewer appointments, days are emptier and you have more time to keep glycemia under control. Even just to pass the time, you put more effort into it. However, it can’t be denied that having to stay home doesn’t help therapy much.*


In our sample, adolescents with T1D did not consider themselves at higher risk for a severe clinical course of COVID-19, when adhering to restrictions. Being considered “frail” was negatively perceived by some participants (5), while others (4) highlighted its advantages (i.e., receiving COVID-19 vaccination earlier during the pandemic).

In our cohort, concerns about getting sick with COVID-19 were rarely disclosed while loved ones’ health (i.e., grandparents) and the possibility of transmitting COVID-19 worried 66% of adolescents.

In no patients, the COVID-19 outbreak led to greater attention to one’s health.

#### 3.2.2. Emotional Reactions to the COVID-19 Pandemic

This theme explores adolescents’ emotional experience during the COVID-19 pandemic, mainly between February 2020 and April 2021.

Most adolescents (78%) described intense and difficult feelings, commonly including sadness, loneliness, frustration, anger, worry, and/or emptiness. Intense and difficult feelings were basically attributed to home confinement, to the inability to maintain usual activities and relationships.


**
*E. (female, 18 years old):*
**
*[...] During the lockdown, I cried many times, for a long time. It had never happened to me before. I was stressed and shut at home. I felt much more lonely.*



**
*A. (female, 18 years old):*
**
*[...] I was scared that something bad could happen to my grandparents. During the first lockdown, people died alone. I was very worried that this would happen to them.*


Symptoms of emotional numbness (i.e., feeling detached from others, feeling emotionally flat) are traceable in the narratives of two male adolescents. 


**
*L. (male, 17 years old):*
**
*[...] It was a very stressful time. I found that, I don’t like to say that, I can be very mean. Some bad things happened and I couldn’t be sad, I didn’t feel anything at that moment. I didn’t feel anything.*


The emotional state of the participants changed over time, worsening between the first (February–March 2020) and the second pandemic wave (October–November 2020), with an improvement in the summer of 2020 and in the months preceding the diabetes camp (July 2021), widely considered the best of the last year and a half, given the recovery of social life. 

In our sample, the experience of intense and difficult emotions appears to be more frequent among older adolescents. A positive adaptation emerges among male participants with six adolescents disclosing that they mostly felt okay during the first year and a half of the COVID-19 pandemic.

#### 3.2.3. Changes in Daily Life

Most of the participants (56%) referred to substantial changes in daily life, including being home confined, having parents more present, taking new necessary precautions (i.e., having to wear masks, social distancing, need to pay more attention), and remote learning. Participants (29%) considered online school boring and stressful. Adolescents were disappointed by too much homework and lack of compassion for their concerns and emotions related to the COVID-19 outbreak.


**
*E. (female, 18 years old):*
**
*[...] Teachers weren’t that kind. School decidedly put extra pressure on me.*


Two participants disclosed modifications in sleep/wake rhythm to ensure themselves greater privacy in the shared domestic space.

Participants (7) referred to the quarantine period as “boring” and “burdensome” with days blurring together and few specific memories to mark time.

#### 3.2.4. Feelings of Loss

Almost all the participants shared feelings of loss. Specifically, 78% of adolescents reported losing leisure activities and experiences such as spending time with family and friends, traveling, or “being just an adolescent”. 


**
*R. (male, 16 years old):*
**
*[...] During the last year, I should have grown, changed, and had certain experiences, but I think I have only grown physically. I found myself growing up as I had never considered before.*


Some participants (22%, mostly female adolescents) highlighted the weakening or loss of some friendship relationships, with a better selection of the truest friends. Loss of physical closeness was experienced by 33% of patients (mostly male participants).

Broader feelings of loss such as loss of freedom, motivation, interest, privacy, or time were furthermore described. 


**
*F. (male, 18 years old):*
**
*[...] What bothers me the most is the time we lost. It is still one and a half years of things we could have done but didn’t!*


#### 3.2.5. Coping with the COVID-19 Pandemic

Participants (79%) disclosed a conscious effort to cheer themselves up and make the most of the situation. Self-care and coping strategies adopted included: learning new things, trying to maintain a positive attitude (e.g., focusing on things to be grateful for), spending time in pleasant activities (i.e., spending time with pets, cooking together, playing games), or regularly exercising. 

For most participants (54%), keeping in touch with friends and family through technology made the first year of the COVID-19 pandemic easier to cope with. However, technology was only considered partially satisfactory, not replacing in-person interactions, thus being associated for some participants (5) with feeling in any way socially disconnected or with the conscious decision to limit distance interactions.


**
*F. (male, 18 years old):*
**
*[...] I love physical contact. I like to hug my friends, stay close, and laugh together. I don’t like to hear my friends on the phone. Even during quarantine, I didn’t use the phone that much because hearing my friends only through messages bothered me. Thus, I closed myself a bit off.*


Coping with the situation became more difficult with time, particularly during the second wave of COVID-19 (October 2020), when the adoption of differentiated containment measures based on the epidemiological risk of the areas required adaptation to rapidly changing life contexts. 


**
*R. (male, 16 years old):*
**
*[...] After the first quarantine, I was quite hopeful. It was summer so with the sun, hot weather, and longer days, it was easier to stay positive. This year, I think I only went ahead out of inertia because we spent every other week locked up at home. There wasn’t even time to get used to and figure out what was happening next.*


#### 3.2.6. COVID-19 Pandemic as Opportunity

Most adolescents (79%) identified some positives in the COVID-19 pandemic. This included more time to self-dedicate, opportunities for self-exploration, and personal growth (37%). 


**
*G. (male, 17 years old):*
**
*[...] I tried to get to know and improve myself. I learned to feel good about myself, without needing someone.*


After home confinement and social isolation, adolescents (32%) reported more appreciating moments previously considered “normal” (i.e., spending time with loved ones), expressing their intention of fully enjoying them once they returned to normality.


**
*E. (female, 15 years old):*
**
*[...] Something that used to be normal, that we underestimated and took for granted, like being together or spending the whole day together, has no longer become something obvious. When I was at home, I learned that I have to enjoy the moments fully.*


Participants (48%) highlighted how lockdowns enabled them to strengthen and fully appreciate their relationships, mainly within their household given more quality time spent together.


**
*B. (male, 16 years old):*
**
*[...] Before COVID-19, I saw my dad for a few hours a day. Now that he spends more time at home, I have gotten to know him better. Let’s say I discovered his “pixelated” side, he was always so precise for his work!*


#### 3.2.7. Return to (New) Normality

This theme describes how adolescents experienced their return to social life.

When asked what they first did once restrictions were eased, 63% of participants disclosed meeting their friends while 29% of our cohort met family members, mostly grandparents. Most adolescents (75%) really enjoyed the first moments back together.


**
*D. (male, 17 years old):*
**
*[...] The first time I met my friends again was awesome! We went together to the Euganian hills, we talked and joked. It was an emotionally intense day. We were still there together, even after being apart for a long time.*



**
*C. (female, 18 years):*
**
*[...] When my grandma hugged me again for the first time, I felt at home. I felt great, I missed her so much.*


Recovering for “lost time” by spending full days together was common among adolescents.


**
*R. (male, 16 years old):*
**
*[...] We spent two full days together, going out all day, even in the evening. We made up for the lost time. It was great!*


Only in a few cases, first social occasions were experienced with initial concern and discomfort.


**
*R. (male, 16 years old):*
**
*[...] At first, it was strange and I felt anxious, not for fear of getting sick with COVID-19, but because I hadn’t had serious contact for months, except by phone. It seemed strange to me, a bit unnatural at first, but then the feeling of anxiety passed and it was a really good moment.*


Despite initial enthusiasm, participants (25%) expressed disappointment in the “new normality” where restrictions, although eased, and the virus spread, continued to condition the way of being together, feasible experiences, or the possibility of fully enjoying them.


**
*M. (male, 17 years old):*
**
*[...] I went to the sushi restaurant with my friends and due to restrictions we couldn’t sit at the same table and we had to split up. It was strange: we were happy to do this again together and then we were separated.*



**
*E. (female, 17 years old):*
**
*[...] My parents showed me photos of trips they took with friends when they were almost my age. Now you can travel, but it is not the same. Everything is more complicated! I think it’s bad! With all those restrictions and checks, I don’t feel like going, honestly, and it seems strange to me because kids should spend their summer differently.*


## 4. Discussion

This study aimed to explore the experience of the COVID-19 outbreak in a cohort of adolescents with type 1 diabetes since previous quantitative studies highlighted better glycemic outcomes [18,19,20,21], but more frequent symptoms of psychological distress [16,18].

Word frequency analysis identified “friends”, “family” and “home” as the most frequent words in patients’ narratives, reasonably summarizing the most salient issues in adolescents’ experience: missing their friends, being stuck at home, more family time, being back together. 

Themes highlighted replicate findings of previous qualitative studies investigating non-diabetic adolescents’ experience during the pandemic [11,12,13,14,15], collectively capturing how the COVID-19 pandemic was an intense and negative experience for youth, with unexpected changes in routines, difficult emotions, and demanding sacrifices. Loss, mainly of milestones and daily adolescent experiences, is among the most recurrent issues in our patients’ narratives along with participants having identified positive opportunities during the COVID-19 outbreak. This seems to suggest that, despite difficulties, for adolescents who have succeeded in an adaptive pandemic appraisal and accommodative coping strategies adoption (i.e., acceptance, compassion, reprioritizing goals), the COVID-19 outbreak resulted in opportunities for self-discovery, self-care, and more intimate relationships. Keeping in touch with friends and family, engaging in pleasant activities, quality family time, and positive thinking helped adolescents make the best of the situation. 

In our cohort, ambivalent feelings characterized the return to normality. While participants enjoyed returning to social life, disappointment with the “new normality” where restrictions and virus circulation continued to condition adolescents’ experiences and time spent together arose. Besides the initial enthusiasm, the return to normality seems to have been actually more complex. In our experience, this speculation is supported by a consistent increase in psychological support requests among adolescents followed at the Pediatric Diabetes Unit (Padua) after restriction easing.

When comparing themes highlighted for our sample with topics that emerged in existing qualitative investigations involving healthy adolescents [19,20,21,22,23], no particular thematic focuses emerged among our participants. Thus, type 1 diabetes seems not to have determined the perception of increased health risk, greater isolation, or stricter adherence to restrictions, replicating the findings of Zeiler et al. [17].

When considering the specificities of our cohort, our investigation identifies aspects that, according to adolescents, influenced adherence to treatment and glycemic control, confirming some speculations in diabetes literature early in the pandemic [19,20,21,22,23]. Slowed-down routines and more time for self-care were associated with better glycemic control. Having to repeatedly adjust T1D therapy to different phases of the emergency, with variable activities allowed, was associated with suboptimal diabetes-related outcomes. Moreover, worse glycemic metrics were attributed to a loss of motivation to self-care, given the circumstances, consistent with the evidence of suboptimal glycemic control among patients with worse mental health [5]. In our patients, there is no evidence of greater involvement of caregivers in T1D management during the emergency. 

Widespread intense and difficult emotions emerged in our cohort. Although such emotional reactions could be normal under the circumstances and prove temporary, action would be desirable. In our experience, narrating their experience within the peer group helped adolescents to normalize worries and emotions experienced in response to drastic and abrupt changes in their developmental context during the COVID-19 pandemic, promoting support perception among patients. Furthermore, participation in focus groups allowed the health care team to immediately intercept discomfort, but also a cohort of patients, sensitized to the importance of mental health care, to directly ask for support and report their symptoms. This had consistent implications on treatment motivation. We find it interesting that 21% of campers autonomously requested psychological support in the months following participation in focus groups.

### Strengths and Limitations

To the best of our knowledge, this study is among the first qualitative investigations exploring adolescents with T1D experience during the COVID-19 pandemic, not limiting itself to the possible impact of the emergency on T1D management and considering a broad timeframe never explored to date.

However, our research has some limitations, mostly concerning sample composition and the moment of data collection. Our small-size sample is gender-inhomogeneous, mostly including male participants. Most participants were experienced in diabetes management, having been diagnosed for more than five years. We hypothesize that aspects more closely related to the impact of the COVID-19 pandemic on T1D may emerge by including more patients with recent diabetes onset.

Two different psychologists, with potentially different management styles, led focus groups. Inevitably, different group dynamics arose between participants. Despite the effort of the moderators to involve all participants, the experience of more introverted adolescents may have been less represented. Similarly, adolescents who had a more difficult time taking care of themselves during the COVID-19 pandemic may have had less intention to share their experiences. Narratives related to the initial period of the COVID-19 emergency may have been affected by the time interval between the pandemic onset (February 2020) and the moment of data collection (July 2021), given the difficulty in marking the pandemic period, highlighted by the participants themselves. Focus groups were held during the evening hours of the diabetes camp; thus, participants could have been tired and unfocused.

## 5. Conclusions

Taken together, our data suggest that the COVID-19 pandemic may have represented a more stressful condition for adolescents with type 1 diabetes, given the need to maintain adequate glycemic control, in the context of sudden and drastic changes in routines, in addition to stressors already affecting the general population of adolescents (i.e., social isolation, remote learning). Being sensitive to mental health importance, easy access to psychological support, and some type 1 diabetes management skills (i.e., planning, frustration tolerance, dealing with the unexpected) may have been protective and facilitated our cohort in stress management and adaptation to the emergency. 

Our results suggest the importance of supporting youth in adaptive appraisal and accommodative coping strategies adoption during stressful events. Findings provide directions to the pediatric diabetes team to promote adolescents’ well-being and to support pediatric patients in reaching adequate glycemic control while the consequences of the COVID-19 pandemic on youth health continue.

## Figures and Tables

**Figure 1 ijerph-20-02620-f001:**
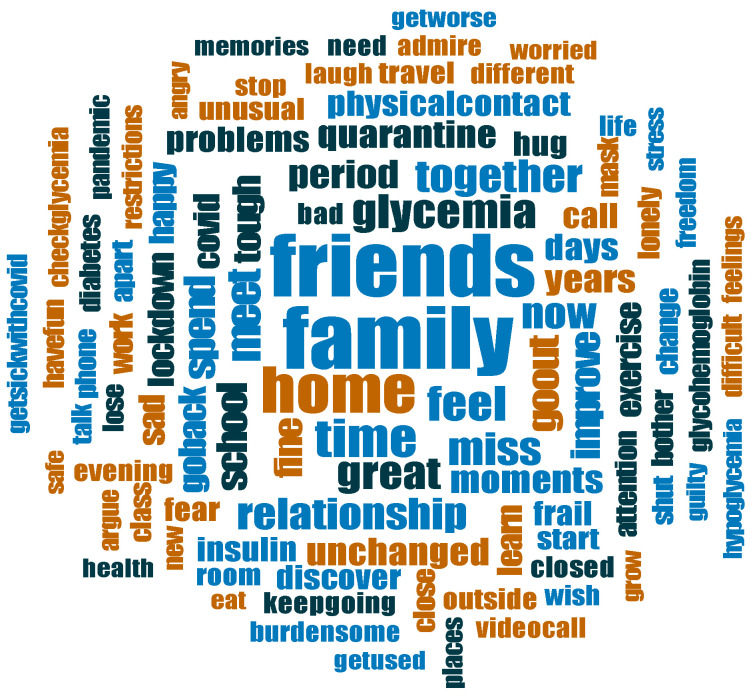
Top 100 most referred words in adolescents’ narratives.

**Figure 2 ijerph-20-02620-f002:**
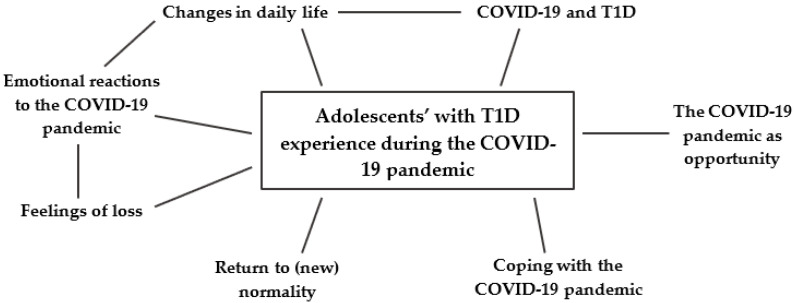
Themes highlighted and relative connections.

**Table 1 ijerph-20-02620-t001:** Demographic and clinical characteristics of the participants at enrollment.

Participants’ Characteristics at Enrollment		Number of Participants
**Age**	15–16 years old	9 (37.50%)
	17–18 years old	15 (62.50%)
**Gender**	male	17 (70.83%)
	female	7 (29.17%)
**T1D duration**	<5 years	7 (29.17%)
	5–10 years	10 (41.67%)
	>10 years	7 (29.17%)
**Therapeutic regimen**	CSII	5 (20.83%)
	MDI	19 (79.17%)

## Data Availability

The data presented in this study are available on request from the corresponding author.

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
