# Peer review of "“One and a Half Years of Things We Could Have Done”: Multi-Method Analysis of the Narratives of Adolescents with Type 1 Diabetes during the COVID-19 Pandemic"

_ijerph, 2023, doi:10.3390/ijerph20032620_

Round 1

Reviewer 1 Report

Τhe article is not bad but it certainly could be better especially if a larger population was included. Τhe main weakness is beyond the small sample, the fact that the sample is heterogeneous which can lead to erroneous conclusionsI realise, however, this is a study that has now been completed and certainly would not have the same value if the sample were to be increased afterwards, given that the emotional burden experienced by the patients during the isolation may have subsided. Of course you mentioned this weakness and I think that with this in mind the study can be read and lead to the appropriate conclusions.

Author Response

D

Dear Reviewer,

thank you for your comments and suggestions. We proceeded to explicit the small size of our cohort on page 10 (“Our small-size sample is gender-inhomogeneous, mostly including male participants”).

We also changed some too much strong interpretation on results in the discussion section.

We hope that, despite the small sample size, our study may still bring to some interesting conclusions.

Reviewer 2 Report

Tremolada and colleagues present interesting fundings of the qualitative research study. There are some places in the manuscript which should be clarified:

Authors start the abstract with the following text “The global prevalence of anxiety and depressive symptoms in adolescents has increased considerably during the COVID-19 pandemic. Mental health problems may compromise glycemic control in young people with type 1 diabetes; however, evidence of improved glycemic control in adolescents with T1D appeared early during the pandemic”. 

Readers could think that authors aim to investigate the incidence of psychiatric symptoms among T1D patients, however this topic was not the case. The goal was “This qualitative
study aimed to provide a more in-depth understanding of how the COVID-19 pandemic affected adolescents with type 1 diabetes routines, experiences, T1D management, behaviors, and mental health”. Rather authors analyzed  experiences and opinions of young people with T1D.

I would recommend to pay a little bit more attention by using terms like depression and anxiety in this context Finally, many works have shown that young people were stressed and lonely, but only very few works could show the clinical depression diagnosis increase during the COVID-19.

Another point authors should shortly discussed is following: The study was done in Italy (Italian young people). This is not a Pandemic what caused loneliness but this were laws/rules of Government who tried to save many people (but probably, and we know it with status today)- this maybe was not really the best way to let elderly people to die alone.  Maybe in the future, Government should not only pretend elderly from death but also young people from psychical traumas.

In Conclusion authors write “Our findings confirm the influence of psychosocial factors on the management of T1D”.  Do they confirm it?  Can we see in this study that psychosocial factors positively improved blood sugar values and hypoglycemia?

Author Response

1) Tremolada and colleagues present interesting fundings of the qualitative research study. There are some places in the manuscript which should be clarified:

Authors start the abstract with the following text “The global prevalence of anxiety and depressive symptoms in adolescents has increased considerably during the COVID-19 pandemic. Mental health problems may compromise glycemic control in young people with type 1 diabetes; however, evidence of improved glycemic control in adolescents with T1D appeared early during the pandemic”.

Readers could think that authors aim to investigate the incidence of psychiatric symptoms among T1D patients, however this topic was not the case. The goal was “This qualitative study aimed to provide a more in-depth understanding of how the COVID-19 pandemic affected adolescents with type 1 diabetes routines, experiences, T1D management, behaviors, and mental health”. Rather authors analyzed  experiences and opinions of young people with T1D.

1 Answer:

Dear Reviewer,

thank you for your comments and suggestions. We proceeded to remove potentially misleading references to our study aims in the Abstract (“Public health interventions for COVID-19 forced families to adopt changes in daily routines that affected children’s and adolescents’ psychological well-being. In youth with type 1 diabetes (T1D), psychological symptoms may compromise glycemic control and outcomes; however, evidence of improved glycemic control in children and adolescents with T1D early emerged during the pandemic. This qualitative study aims to provide a more in-depth understanding of how the COVID-19 pandemic affected adolescents with T1D routines, experiences, T1D management, and psychological well-being.”).

2. I would recommend to pay a little bit more attention by using terms like depression and anxiety in this context Finally, many works have shown that young people were stressed and lonely, but only very few works could show the clinical depression diagnosis increase during the COVID-19.

2 Answer

We removed improper references to psychiatric symptoms preferring expressions such as “emotional distress”, “psychological well-being deteriorations” and “more frequently emotionally distressed”.

3. Another point authors should shortly discussed is following: The study was done in Italy (Italian young people). This is not a Pandemic what caused loneliness but this were laws/rules of Government who tried to save many people (but probably, and we know it with status today)- this maybe was not really the best way to let elderly people to die alone.  Maybe in the future, Government should not only pretend elderly from death but also young people from psychical traumas.

3 Answer

We proceeded to better explicit the relationship between psychological distress and public health interventions during the pandemic in the Abstract (“Public health interventions for COVID-19 forced families to adopt changes in daily routines that affected children’s and adolescents’ psychological well-being”) and on page 2 (“During the COVID-19 pandemic, public health interventions against COVID-19, such as “stay at home” order, school closures, and limited peer interactions, have taken their toll on adolescents’ psychological well-being, resulting in a consistent increase in reported symptoms of emotional distress”).

4. In Conclusion authors write “Our findings confirm the influence of psychosocial factors on the management of T1D”.  Do they confirm it?  Can we see in this study that psychosocial factors positively improved blood sugar values and hypoglycemia?

4 answer

On page 10, we removed misleading references to the association between psychosocial factors and T1D management and tried to better describe our study implications.

With these improvements and integrations, we hope our study can be considered interesting to be shared.

Reviewer 3 Report

Overall, the manuscript submitted for review focuses on an important and under-researched issue. The study and identification of determinants of glycemic control in adolescents with type 1 diabetes, especially during the COVID-19 pandemic, is very valuable and needed.  The research undertaken is therefore a response to the need in this area.

Article is well-organised. The layout of the article is typical of original papers and includes all required parts. Some sections are well developed, but some require improvement (e.g. results and discussion).

In section 1 “Introduction” the literature data has been well summarized, but in Section 4, "Discussion," the comparison of the authors' results with those of other authors needs to be supplemented and more current literature used. References to literature sources are not given after most paragraphs in “Discussion” section. This requires supplementation (paragraphs 3, 4, 5 after the words "in previous qualitative studies"; 6 after the words "in the diabetes literature …" and at the end of this paragraph).

All aspects of the methodology are clearly explained.

In the "Results" section, the removal of quotes should be considered. The results of the issues analyzed, taking into account the responses of all survey participants, not just selected individuals, should be grouped and presented in tabular form, for example. The authors in many places state that a given result was recorded in, for example, 54%, 32%, 48% or 29% of the respondents and provide only one or two citations. This does not represent all the results obtained.

In some places in the description of the results, the authors only state, for example, "most participants" or only "participants" without specifying the percentage of respondents. This needs to be clarified.

The Authors achieved their goals, which allowed them to draw conclusions, and accurately identified some limitations and strengths of the study. In the limitations, the authors should take into account the small size of the group.

Other comments include:

1)      Should the title use the abbreviation type 1 diabetes? In my opinion, the full name of the disease should be given.

2)      All acronyms should be defined the first time they appear in each of two sections: the abstract (T1D, DM1) and the main text.

3)      The “Results” section (last paragraph) lacks explanations of the abbreviations M or SD with COVID-19 and T1D.

4)      T1D or DM1? - Decide on one disease abbreviation.

5)      DD1 - what does the abbreviation entered in the conclusions mean?

6)      The conclusions should include the full name of the disease.

7)      A.N and M.C - is it necessary to give the initials of the psychologists?

8)      A.D. and M.T. - is it necessary to give the initials of the researches?

9)      In the keywords, instead of NVivo, you can specify the Nvivo program.

10)  In the abstract, it is worth mentioning the names of the tools that were used for the analysis.

1     11)  In the “Materials and Methods” section, please complete the ethics committee approval number.

12)  In abstract: “Word Frequency Analysis” or “word frequency analysis”

Author Response

Article is well-organised. The layout of the article is typical of original papers and includes all required parts. Some sections are well developed, but some require improvement (e.g. results and discussion).

1. In section 1 “Introduction” the literature data has been well summarized, but in Section 4, "Discussion," the comparison of the authors' results with those of other authors needs to be supplemented and more current literature used. References to literature sources are not given after most paragraphs in “Discussion” section. This requires supplementation (paragraphs 3, 4, 5 after the words "in previous qualitative studies"; 6 after the words "in the diabetes literature …" and at the end of this paragraph).

1 Answer

Dear Reviewer,

thank you for your comments and suggestions. We tried to improve “Discussion” and “Results” sections. On page 2, we tried to supplement our results with those of existing qualitative evidence (“In qualitative research having investigated adolescents’ experience during the COVID-19 outbreak, difficult emotions (e.g. sadness, anger, worry, irritability), home confinement, loss of milestones and lack of peer interactions are common issues”).

We added references to literature sources in “Discussion” section.

2. All aspects of the methodology are clearly explained.

In the "Results" section, the removal of quotes should be considered. The results of the issues analyzed, taking into account the responses of all survey participants, not just selected individuals, should be grouped and presented in tabular form, for example. The authors in many places state that a given result was recorded in, for example, 54%, 32%, 48% or 29% of the respondents and provide only one or two citations. This does not represent all the results obtained.

2 Answer

On page 5 we tried to better explicit our intention to only include participants’ most relevant and exhaustive quotes for each theme (“A complete description of the main and sub-theme, including participants’ most relevant and exhaustive quotes, is presented below”) which, in our opinion, add value to the research.

I3. n some places in the description of the results, the authors only state, for example, "most participants" or only "participants" without specifying the percentage of respondents. This needs to be clarified.

Answer 3

We proceeded to specify the percentage of participants who introduced each sub-theme.

4. The Authors achieved their goals, which allowed them to draw conclusions, and accurately identified some limitations and strengths of the study. In the limitations, the authors should take into account the small size of the group.

4 Answer

We proceeded to explicit the small size of our cohort on page 10 (“Our small-size sample is gender-inhomogeneous, mostly including male participants”).

Other comments include:

1)      Should the title use the abbreviation type 1 diabetes? In my opinion, the full name of the disease should be given.

We inserted the full name of the diseases (type 1 diabetes) in the title.

2)      All acronyms should be defined the first time they appear in each of two sections: the abstract (T1D, DM1) and the main text.

We proceeded to define all acronyms the first time they appear.

3)      The “Results” section (last paragraph) lacks explanations of the abbreviations M or SD with COVID-19 and T1D.

At the end of page 4 we reformulated better the sentence to understand that M stays for Mean and SD stays for Standard Deviation. We didn’t add also the definition of M and SD that is universally known, but this reformulation makes all more clear. Thank you for suggestion.

4)      T1D or DM1? - Decide on one disease abbreviation.

We have chosen T1D.

5)      DD1 - what does the abbreviation entered in the conclusions mean?

We are sorry, it was a typing error. It should have been T1D.

6)      The conclusions should include the full name of the disease.

Conclusions now include the full name of the disease.

9)      In the keywords, instead of NVivo, you can specify the Nvivo program.

In the keywords, we specified “NVivo program”.

10)  In the abstract, it is worth mentioning the names of the tools that were used for the analysis.

We add “Descriptive analyses and t-test were run to answer the research questions” in the abstract.

11)  In the “Materials and Methods” section, please complete the ethics committee approval number.

At the end of page 3 we added “This human research had a negligible risk and it belong to the ordinary clinical assistance activity. For these reasons it could apply for an exemption from ethics review”. The term ‘negligible risk’ (as taken from the National Statement on Ethical Conduct in Human Research (Chapter 2.1, Paragraph 2.1.7)) describes research in which there is no foreseeable risk of harm or discomfort; and any foreseeable risk involves no more than inconvenience.

12)  In abstract: “Word Frequency Analysis” or “word frequency analysis”

We have chosen “word frequency analysis”.

With these improvements and integrations, we hope our findings can be considered interesting to be shared.

Round 2

Reviewer 2 Report

No further comments